# An automatic music generation and evaluation method based on transfer learning

**Yi Guo, Yangcheng Liu, Ting Zhou ⓘ\*, Liang Xu, Qianxue Zhang**

Xihua University, Chengdu, China

\* luming9704@163.com

## Abstract

In recent years, deep learning has seen remarkable progress in many fields, especially with many excellent pre-training models emerged in Natural Language Processing(NLP). However, these pre-training models can not be used directly in music generation tasks due to the different representations between music symbols and text. Compared with the traditional presentation method of music melody that only includes the pitch relationship between single notes, the text-like representation method proposed in this paper contains more melody information, including pitch, rhythm and pauses, which expresses the melody in a form similar to text and makes it possible to use existing pre-training models in symbolic melody generation. In this paper, based on the generative pre-training-2(GPT-2) text generation model and transfer learning we propose MT-GPT-2(music textual GPT-2) model that is used in music melody generation. Then, a symbolic music evaluation method(MEM) is proposed through the combination of mathematical statistics, music theory knowledge and signal processing methods, which is more objective than the manual evaluation method. Based on this evaluation method and music theories, the music generation model in this paper are compared with other models (such as long short-term memory (LSTM) model,Leak-GAN model and Music SketchNet). The results show that the melody generated by the proposed model is closer to real music.

**Data Availability Statement:** All midi files are available from the POP909 database, Wang Z, Chen K, Jiang J, et al. Pop909: A pop-song dataset for music arrangement generation[J]. arXiv

## Introduction

Music is everywhere in our lives. The rapid development of all walks of life has boosted the demand for music. The task of automatic music generation has attracted widespread attention. Automatic music generation methods can be divided into symbolic music generation and audio music generation. In this paper, we mainly study the generation of symbolic melody in symbolic music generation.

Most music categories, such as pop music, are composed of melody and harmony. The melody is the key to determine whether the music is good or not, and it is a linear continuous signal. The melody has information such as rhythm and pause, which makes the melody generation a challenging task. With the rising deep learning [1], various types of neural network models have been used in automatic melody generation. In some recent studies, such as [2–4], the use of neural network model greatly simplifies the problem of melody generation. Some

preprint: 2008.07142, 2020. (https://doi.org/10.48550/arXiv.2008.07142).

**Funding:** Xihua University Graduate Innovation Fund, ycjj2020118, YL, Intelligent Terminal Key Laboratory of SiChuan Province, SCITLAB-1021, YG, the National Natural Science Foundation of China under Grantï61973257, 61901394ï YG.

**Competing interests:** The authors have declared that no competing interests exist.

methods such as [5] generates melody and rhythm separately. Such representation method can not well capture the characteristics of the melody.

Refer to the way the text is generated, in theory, we can combine the melody pitch of the notes with the melody rhythm, and express musical melody in a form of text, so as to integrate the pitch, rhythm and other information into the text.

At the same time, we found that there are various problems in the existing melody generation. For example, in LSTM model [6], the length of the learnable melody is relatively large, and it is usually impossible to generate long-term music melody. In order to solve this problem, we noticed several large-scale pre-training models that work well in text generation (such as BERT [7], GPT-2 [8], etc.), but there is a data representation problem when apply these pre-training models in automatic music generation. Due to the expression limitation of musical melodies, the musical melody expression method we proposed above comes in handy.

We propose a textual music melody generation method based on GPT-2 model, which transfers the large-scale pre-training model GPT-2 to the task of music generation, and it can generate long-term time series melody and simplify the music generation. And this method can also be quickly transplanted to other large text generation model, so that it can be trained and generated quickly.

Secondly, objective music melody evaluation methods have always been a gap in automatic music generation. For example, in [3, 5, 9], the evaluation heavily relies on people's feedback. However, the people's different perception of music makes such evaluation methods less practical and objective in reality. In another way, some papers such as [10] only analyzed the accuracy of the notes, and scarcely made any evaluation on the musicality of the melody. Some evaluation methods of mathematical statistics are used in [11], which are still imperfect. Therefore, there is an urgent need for a more objective and comprehensive evaluation method.

In order to solve this problem, we propose a music evaluation method that combines mathematical statistics and music theory knowledge to objectively evaluate the generated music melody through the degree of note change in the melody and the wavy nature of the melody. The results show that the melody generated by textual music melody generation method based on GPT-2 is closer to real music than LSTM model [6],Leak-GAN model [12] and Music SketchNet [13].

## Related work

### Melody generation with neural networks

Many researchers have used the traditional probabilistic generation method for automatic music generation, mainly including N-gram or Markov model [9, 14, 15]. Later, Bettan et al. [16] proposed a music fragment selection method, which create as new music by calculating the similarity ranking between music fragments. On the other hand, Pachet et al. [17] used chords to select melodies. These methods of establishing probability model are feasible to some extent but limited. First, because of the diversity and development of music, the probability model cannot be updated in time. Second, to establish a good probability model requires deep knowledge of music theory. Moreover, traditional methods need to design and extract a large number of manual features, it takes a lot of manpower and time.

With the development of deep learning in recent years, deep neural network has been applied to automatic music generation, which has solved part of the above problems. Franklin [18] proposed the possibility of using recurrent neural network(RNN) to represent multiple notes at the same time, so as to generate more complex music sequences, Goel [19] proposed a polyphonic music generation method based on RNN, and this model generates multiple parts by using Gibbs sampling method. Different from the RNN model, Sabathe et al. [4] used

variational auto-encoder(VAEs) to learn the distribution of music fragments. In addition, Yangetal used generative adversarial networks(GANs) [20] to generate music and used random noise as input to regenerate melody. Although a great deal of work has been done on music generation, many musical elements have not been fully considered, such as note duration, note pause, music style, music format of input model, etc. Due to the differences in the representation of music signal and text, the pre-training model of text generation has not been transfered to music generation. Also, the advantage of deep learning in automatic music generation is limited by the scarcity of good music data sets.

The preprocessing methods for music data are not unified and standardized. In music generation based on note representing, the processing method of musical notes is relatively difficult, and the melody pitch (such as C4, C4, G4, G4 and others) is directly extracted, neglecting the rhythm of melody (duration of notes, pause, etc.). For example, in [3, 5], Hongyuan Zhu proposed the model of cross generation of melody and rhythm (training pitch and rhythm separately), although taking into account pitch and rhythm, it is still not compatible in training and generation. such as the interactive melody generation method based on the VAEs [2].The music extraction in [13, 21] uses the customized frame number to extract music, which may cause the loss of music information.

## Music evaluation

Music evaluation method has always been a vacancy in the study of automatic music generation. In some papers, such as [11], volunteers were invited to test the quality of music generation, but everyone has different tastes towards music, this method is rather subjective. It is necessary to establish an objective music evaluation index to assess the quality of the generated music. In some papers, such as [10], only the model accuracy rate (the accuracy of predicting the next note based on previous notes, etc.) was analyzed for the results of music generation. But these tests are not accurate for the generated music, which values "quality" over "quantity" and "new" over "old". Wilcoxon test (signed rank test), Mann-Whitney U test and other mathematical statistical methods, to test whether the data comes from the same population [22]. Most of these existing methods are subjective, and mathematical statistic method is objective but not competent, because they are not integrated with the characteristics of the music itself. Therefore, there is a urgent need for a more objective method with resonable indicators to test the quality of the generated music.

## Music representation

We first briefly introduce some basic forms of music data, which can familiarize the readers who do not have a music background, then explain how to express the music melody.

### Formats of music data

According to three different formats of music, we made the conversion diagram in Fig 1, which describes three main music data formats and the method of conversion to each other. MIDI(musical instrument digital interface) is our main research format, which is created to solve the problem of communication in the electro acoustic musical instrument in the beginning of 1980s. MIDI is the most extensive music standard format in composing, which can be called "computer understandable score". It uses the digital control signal of the notes to record music. A complete MIDI music is only dozens of KB in size and contains dozens of music tracks. Almost all modern music is composed using MIDI as a sound library. MIDI transmits not sound signals, but events such as notes, controling parameters, it tell the MIDI device what to do and how to do it, such as which notes to play, duration, volume, etc.

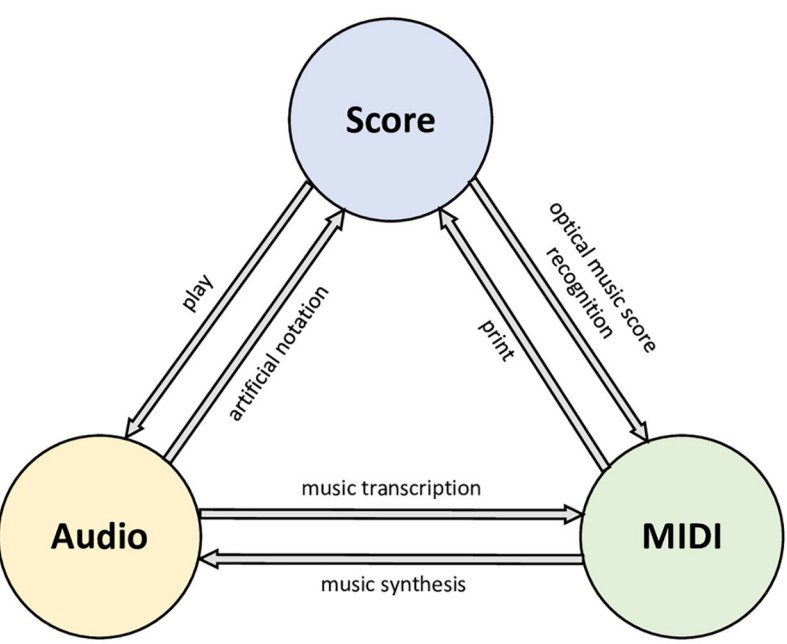

**Fig 1. Representation of music data and its conversion.**

## Melody representation

To simplify the problem, we first extracted the melody sound tracks and represented them by text drawing. In the past traditional music generation rarely notice the notes of delay and pause when the melody was extracted. For example, in Fig 2, the duration of the two notes (The notes are framed in red) is very small, and the direct extraction of pitch information cannot represent the difference between notes, so it is necessary to reduce the unit duration and use a small unit length to represent all MIDI notes. Similarly, the extraction method in Deepbach [23] is rigid and does not consider the pause of notes, which makes the song lacks of rhythmic information. With reference to music theory [24], we propose the pretreatment methods discripted in part IV.

## Model

In this section, we first introduce the data preprocessing, and then the music textual-GPT-2 (MT-GPT-2) model. In the end, we introduce the music evaluation method(MEM). System flowchart of the model is shown in Fig 3.

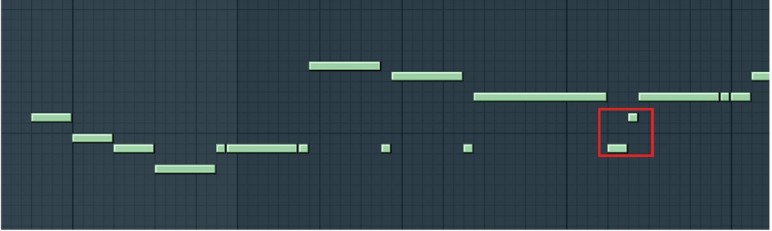

**Fig 2. Example of melody in midi file.** (Use FL Studio DAW for presentation).

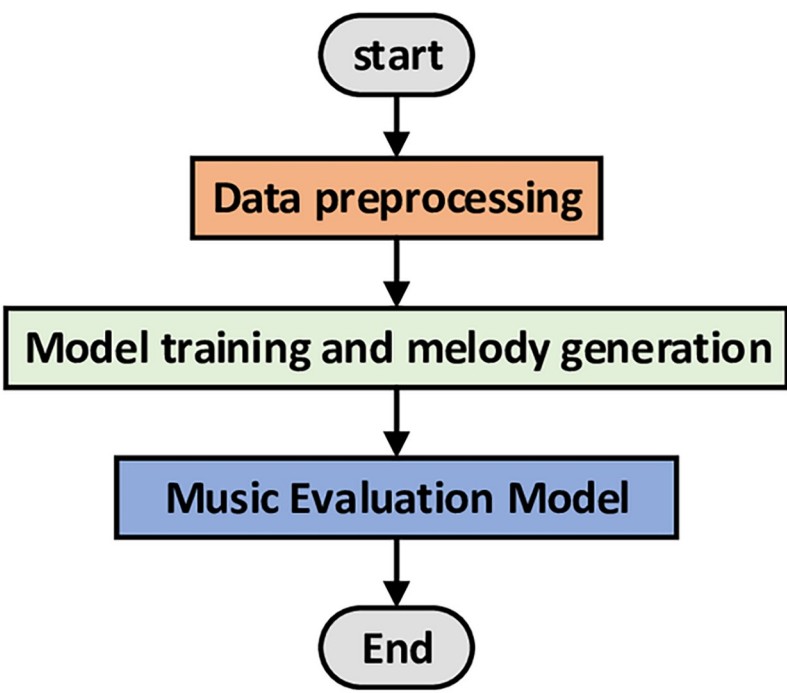

**Fig 3. System flow chart.**

## A novel method of musical melody text representation

The flow of the novel data preprocessing is shown in the Fig 4.

A music textual method is designed to guide the automatic music generation. Converting melody into one-dimensional music text signal makes it possible to apply the text processing method to music without affecting the music information.

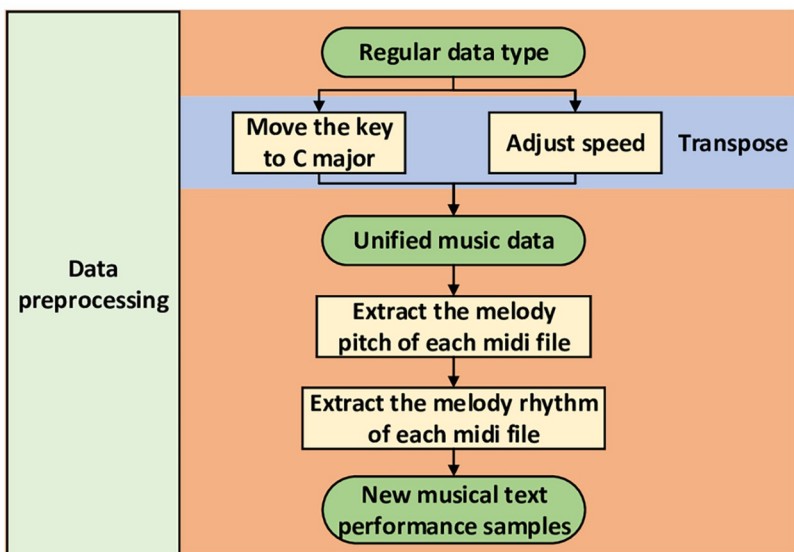

**Fig 4. Flow chart of data preprocessing.**

The preprocessing makes the music representation consistent with the input of most text generation models, which improves the current problems of different models input in automatic music generation and difficulties in model design. We transfer the musical mode and adjust the speed. Secondly, we extract the pitch and rhythm information of the notes to get the musical melody text. The steps are as follows:

**Transpose.** The purpose of this step is to convert all MIDI music to the same mode and the same speed, because such data normalization is beneficial for model learning. First, by calculating the distance between the modality of this MIDI and the key of C, we convert all the notes to the key of C by tonal conversion. After that the tonics of all music files are C, and the rest of the melody changes around the tonic in a regular manner. Then, we set the speed of all MIDI music to 90, which allows all MIDI music to play at the same speed.

**Extract note length information.** We obtain the duration $T_0$ of the shortest note in the melody score, and use $T_0$ as the basic unit length to extract the duration information of all notes, which solves the problem of missing duration information in the conversion process.

The specific steps for extracting note duration information are as follows:

- Specify the MIDI music file and extract the melody track from $Track_1$;

- Get the list by traversing all the notes in it and extracting the duration $L_1$, Get the smallest duration $T_0$;

- After obtaining the duration $T_0$, divide the duration of all notes by $T_0$ to obtain the note multiple $N_i$ of all notes for the smallest note, so as to facilitate the textual representation below.

**Extract pitch and rhythm information.** According to the previously extracted unit length $T_0$ as the minimum length, we can extract notes, note duration, and pause duration at the same time. The extracted sample is shown in Fig 5. For notes, we directly use letters to represent them, starting from the center C of the piano, The center C is called C4. Pushing to the right in turn is C4 D4 E4 F4 G4 A4 B4 (white key), after B4, pushing to the right is C5 D5 E5 F5 G5 A5 B5, where E-4 is the black key between D4 and E4. The key can be represented by E-4 (-: flat note) or D#4 (#: sharp note) that is flattened to E4.

Rhythm information indicates the duration and pause. The note "-" is used here to indicate that the previous note is extended according to our unit time (the shortest note). The "^" symbol is used here to indicate the duration of the pause. In this way, there is not any missing information for the representation of the melody.

The steps of textual music melody are as follows:

- Specify the MIDI music file and extract the melody track from $Track_1$;

- Starting from the first note, as shown in Fig 5 above, extract the pitch of the first note C4;

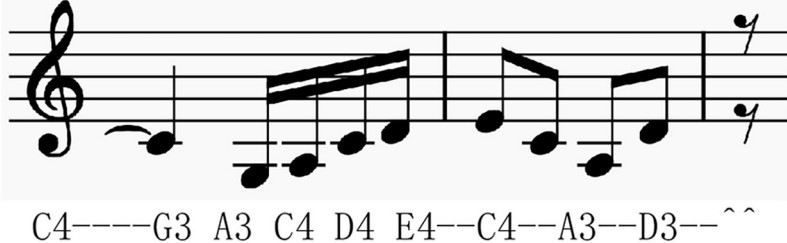

**Fig 5. Example of textual conversion, the graph of the upper part is the musical melody, and the bottom is the corresponding textual melody text.**

- Determine whether the duration information of the note is a unit length. If yes, we start to judge the next note. If it is not a unit length, fill in the duration information based on the unit length information $N_i$ obtained above, and use the symbol "-" to indicate the duration information of the note. For example, the following C4, the obtained $N_i$ is 4 units of duration, add 4 "-" at the end;

- If the next note is a pause, use the pause duration divided by $T_0$ to get the multiple of the unit duration $T_0$. As shown in the figure below, the pause obtained after "D3- -" is 2 units of duration, so fill in two "^" symbols.

## GPT-2 based on music textualization(MT-GPT-2)

For the Large pre-training model, we choose to use the GPT-2 model [8]. based on GPT-2 model and transfer learning we propose MT-GPT-2(music textual GPT-2) model. This section introduces the GPT-2 model and attention mechanism, then introduces the model used and the regulating part of the model.

Large-scale natural language processing models such as BERT [7], Transformer XL [25], and XLNet [26] have taken turns to set records in various natural language processing tasks. Among them, GPT-2 has attracted the attention of the industry due to its stable and excellent performance. GPT-2 has an amazing performance in text generation, and the generated texts exceed people's expectations in terms of contextual coherence and emotional expression. In terms of model architecture alone, GPT-2 does not have a particularly novel architecture. It is very similar to the transformer [27] model with only a decoder, but because of the previous data format problems, it is still lacking in automatic music generation. So transfering it to music generation is worth studying.

The complete block diagram of the GPT-2 model is shown in the Fig 6.

$$P(x) = \prod_{i=1}^{n} P(s_n | s_1, ..., s_{n-1}) \tag{1}$$

This method is convenient to estimate $P(x)$ and $P(s_{n-k}, ..., s_n | s_1, ..., s_{n-k-1})$ for any conditions. In simple terms, it is to use the previous data to predict the following data. A breakthrough of GPT-2 over other pre-training models is that it can perform well in downstream tasks such as reading comprehension, machine translation, question and answer, and text summarization without training for specific downstream tasks. And unsupervised language modeling can learn the features needed for supervised tasks.

For the unsupervised corpus words $U = (u_{i-k}, ..., u_{i-1})$ provided in the data, we use the model object to maximize the following:

$$L_i(u) = \sum_i \log P(u_i | u_{i-k}, ..., u_{i-1}; \theta) \tag{2}$$

In the above formula, $k$ is the size of the context window, and the conditional probability $P$ is modeled by the neural network parameter $\theta$. These parameters were trained using stochastic gradient descent.

In our experiment, we used a multi-layer transformer decoder as a language model, this is a variant of Transformer [25]. This model uses a multi-headed self-attention operation and then passes through the position-wise forward feedback network to generate the output distribution of target words:

$$h_0 = UW_e + W_p \tag{3}$$

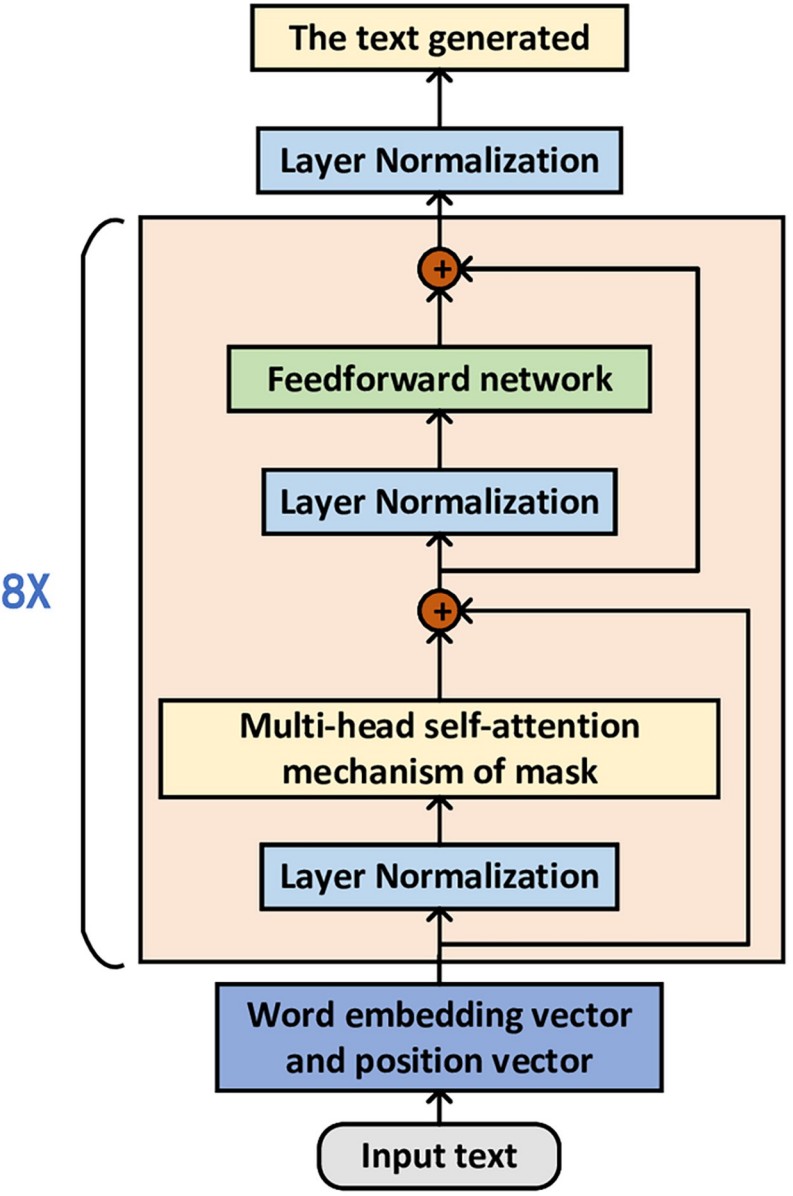

**Fig 6. GPT-2 model flow chart.**

$$h_l = transformerblock(h_{l-1}) \forall l \in [1, n] \tag{4}$$

$$P(u) = softmax(h_n W_e^T) \tag{5}$$

$U = (u_{i-k}, \ldots, u_{i-1})$ is the context vector of the word, $n$ is the number of levels, $W_e$ is the word embedding matrix, and $W_p$ is the position embedding matrix.

Self-attention mechanism is to find the influence of each other words on the current word input in a paragraph. For example, "I have a dog, and it is very good". When dealing with "it", the word "dog" has the greatest impact on "it". The attention mechanism is proposed in the

paper [27], and the self-attention mechanism formula is as follows:

$$Attention(Q, K, V) = softmax\left(\frac{QK^T}{\sqrt{d_k}}\right)V \tag{6}$$

The $Q$ (Query) vector is the representation of the current word and is used to score all other words (using their keys). The $K$ (key) vector is similar to the labels of all words in the segment. They are the objects we want to match when searching for related words. The $V$ (value) vector is the actual word representation.

Masked Self-Attention means that when we process the input sentence and calculate the influence of the attention mechanism, we only pay attention to the current word and the words input before the current word, and mask the words after the word. The advantage is that it can extract the attention weight in one direction, remove the influence of subsequent words, and prevent the occurrence of exactly the same information for generation. This method reduces the accuracy of the model, but in the case of music generation, it avoids producing the same music.

Masked Muilti Self Attention means that when performing matrix operations, the extracted $Q$, $K$, and $V$ vectors are cut and then operated separately, then the results are added. The advantage of this is that each can be extracted more comprehensively attentional influence between words.

The adjustment part is shown in Fig 7.

As shown in Fig 7, the MT-GPT-2 generation model is based on our specific preprocessing method. The construction of a note dictionary is added before the training model. The note is converted to melody text file and extract all the elements in it to make a dictionary, which will be used for model training. And make interface changes to the input and output of the model.

To transfer GPT-2 model to automatic music generation, we use the simplest and most direct method, it is to text the music data, so that GPT-2 model can directly use these data. However, due to the difference between music text and text, the model needs to be adjusted. For the textual data we produced, we constructed a dictionary of notes, then sorted and destressed all the notes that appeared. The Tokenizer method in the BERT model [7] is used to get a note dictionary, and the data set is divided into 100 small data subsets.

Compared with the GPT-2 model, the parameters in the MT-GPT-2 model should be adjusted. First of all, the original dimension of the word vector is set to 1024 that is too large

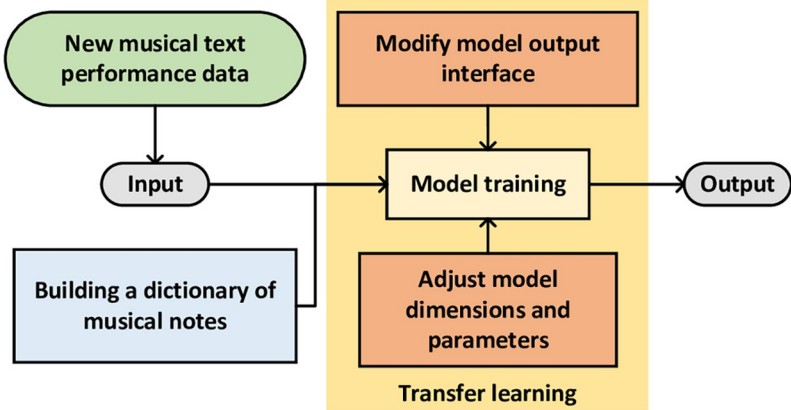

**Fig 7. GPT-2 model fine-tuning.**

for musical notes, which is not much for articles, but too large for musical notes. There are only 144 musical notes on a piano key, so it is necessary to shrink the dimension. Secondly, we increase the number of heads of attention mechanism, because there are more features of musical melody than text. Increasing the number of heads is necessary to learn the relationships between notes in multiple ways. For each layer in the neural network, we increase the width because the length of a song is longer than that of text, and increasing the width can enhance the model's learning for long musical sequences.

For the output of the model melodies, some adjustments are made to the generated musical melody. Any note or melody fragment can be inputed as driving note, based on which the subsequent notes are generated by the model. The length of the generated notes is set to 1024, so that the duration of the generated music melody is about 150 seconds. The model generates one note at a time, and one of the highest 8 notes is selected, which increases the randomness and innovation of the melody.

## Music evaluation method(MEM)

The framework of the music evaluation model(MEM) is shown in the Fig 8.

Compared with image generation, both of them emphasize "true", while music generation emphasizes "quality" rather than "quantity", and "new" rather than "old". "New" requires the model to be creative instead of repeating the learned segments all the time. The existing evaluation of "quality" is very subjective. In order to ensure the novelty of the generated melody, we evaluate these melodis in terms of mathematical statistics and music theory knowledge [28]. The evaluation model-MEM (Music evaluation model) propoesed in this paper is divided into the following mathematical statistics test and music theory evaluation.

**Mathematical statistics test.** Mathematical statistics [29] is an applied mathematics subject that conducts quantitative research on the results of a limited number of observations or experiments on random phenomena, and makes certain reliable inferences based on the overall quantitative regularity. It is the study of how to effectively collect, organize, and analyze randomized data that aims to make inferences or predictions about the problems under investigation, and it provides a basis and basis for certain decisions and actions.

Combing statistical methods used here and below are concerned, they all study whether the generated music (random sequence $A_n$) belongs to the real music population ($R$) in the data set. Our goal is to see whether the generated music also obeys the same overall distribution. If these melodies follow the same overall distribution, this illustrates the the generated music and the real music are statistically similar from a mathematical point of view. Wilcoxon Test, Mann-Whitney U Test and Kruskal-Wallis H Test were used to test the melodies respectively.

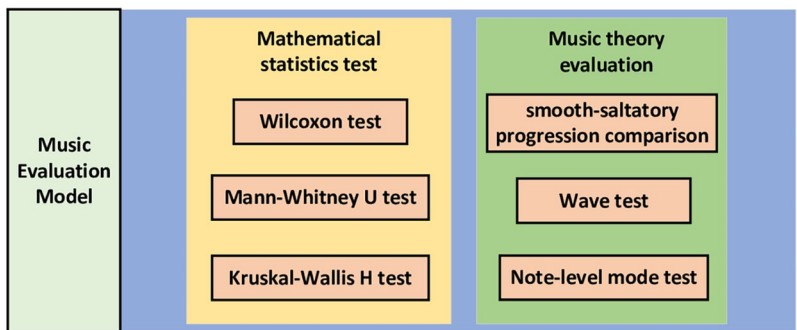

**Fig 8. Block diagram of music evaluation model.**

*i) Wilcoxon test*. In the Wilcoxon signed rank test, it adds the rank of the absolute value of the difference between the observation value and the center position of the null hypothesis. It is suitable for pairwise comparison in the T test, and it does not require the difference between the paired data to obey a normal distribution, but a symmetrical distribution. Test whether the difference between paired observations comes from a population with a mean of 0 (whether the population that produces the data has the same mean).

Both the positive and negative sign test and the Wilcoxon signed rank test can be regarded as substitutes for the parametric T test for paired observations. The nonparametric test has the advantage of not needing to make assumptions about the overall distribution. For the parametric T test of paired observations, it must be assumed that the overall difference is normally distributed.

The steps of the method are as follows:

- We assume $H0$: There is no significant difference between the generated music and the real music. $H1$: There is a significant difference between the generated music and the real music.

- Taking into account the differences in musical modes, which vary greatly from song to song, we subtracted the preceding note from the number corresponding to the pitch of each note to get a sequence $D_i$ that records the variation of the notes.

- We use the Wilcoxon Test function in the scientific computing package in Python to compare the generated melodic sequence $D_1$ with the actual musical sequence $D_2$ to see if there is any significant difference.

- Get the $p$ value.

Calculate the $P$ value, and then specify the significance level $\alpha$ (generally 0.05). When $P > \alpha$, we accept $H0$ and reject $H1$, thinking that the generated music is not significantly different from the real music.

*ii) Mann-Whitney U test*. Mann-Whitney U test, also known as Mann-Whitney rank sum test, can be regarded as the T test of the parameter test method of the difference between two means or the corresponding large sample normal test substitutes. Since the Mann-Whitney rank sum test explicitly considers the rank of each measured value in each sample, it uses more informations than the sign test method.

The steps of the detection method are as follows:

- Similar to the the Wilcoxon Test, we test whether the generated music is significantly different from the real music, that is, hypothesis $H0$: There is no significant difference between the two music, and $H1$: There is a significant difference between the two music.

- Preprocess the obtained music sequence to obtain the corresponding generated music melody sequence $D_1$ and real music melody $D_2$.

- Use the Mann-Whitney U test function in the scientific computing package in Python to get the $U$ value.

Calculate $U$ according to the above steps, and compare with the critical value $U\alpha$ ($\alpha$ is generally 0.05, $U\alpha$ can be obtained by looking up the table), when $U \leq U\alpha$, reject $H1$ and accept $H0$, which is no significant difference between generated music and real music.

*iii) Kruskal-Wallis H test*. In practice, it is often necessary to compare the differences between the means of multiple independent data. Existence problems sometimes affect the results of more than one factor. In this way, it is necessary to carry out combined experiments and repeated sampling of various factors at different levels. When the test error is too large, it

is not conducive to comparing the difference, because too many samples cannot be allowed in a combination.

In addition, it is also necessary to consider that the data in a group should meet the homogeneity. When extract data, it is necessary to consider how to better design the experiment according to the randomness of the data source. Kruskal-Wallis test is a method that extends the Wilcoxon-Mann-Whitney test of two independent samples to three or more group tests.

We assume $H0$: There is no significant difference between the generated music and the real music and $H1$: There is a significant difference between the generated music and the real music.

Calculate the $p$ value through the scientific computing package in Python, and then specify the significance level (generally 0.05). When $p > \alpha$, we accept $H0$ and reject $H1$, thinking that the generated music is not significantly different from the real music.

**Music theory evaluation.** The current evaluation of automatic music generation doesn't take the music theory into account. In other words, it is not comprehensive to the evaluation of only the accuracy of the model and mathematical statistics. Therefore, servral indicators for music evaluation are proposed in this paper, including smooth-saltatory progression comparison, wave test and note-level mode test.

*i) Smooth-saltatory progression comparison.* In terms of music harmony, there is the term "smooth progression". One degree, two degrees and three degrees of progression are called "smooth progression". Four degrees and five degrees or higher degrees are called "saltatory progression".

Progression is a form of melody progress, and it refers to the ascending or descending of a second interval between two adjacent notes in a melody in the order of the scale. For example, there are two notes where the first note is C4 and the second note is D4, the interval between them is called the major second, such a transitional relationship between two notes is referred to in this paper as "smooth progression". If the first note is C4 and the second is F4, the interval between them is a perfect fourth, and we call it the "saltatory progression". The upward direction is generally stronger and stronger, which means to emphasize the theme of the song, while the downward direction is more peaceful.

We perform a digital representation of the generated music, and then perform a difference operation. First we get the note sequence $a_1, a_2, a_3, \ldots, a_n$, where $a_{n+1} = a_n + d (n = 1, 2, 3, \ldots, n)$, $d$ is a constant, and is called tolerance. Therefor, we can get the sequence of $d$:

$$D(a_n) = a_{n+1} - a_n \qquad (7)$$

After obtaining the sequence $D(a_n)$, we pick out the number of values whose absolute value is bigger than 4 (representing four degrees), and compare it with the number of values less than or equal to 4 to get the smooth-saltatory progression ratio:

$$q = \frac{(D(a_n) \geq 4)}{(D(a_n) < 4)} \qquad (8)$$

In addition, the degree of progression of the transformation can also be analyzed through the spectrum after the discrete fourier transform.

*ii) Wave test.* The progression of musical melodies is up and down, and appears to be a positive, flowing waveform in outline. In the progression of the melody the notes always start on a steady note, then go up or down, and finally return to the steady note. Therefore, we detect the waveform in progress of the generated melodies, and then compare them with the waveform of the real melodies to see whether there is an obvious difference, so as to judge the quality of the generated melody.

According to the shape of the melody, it is roughly divided into seven shapes [28].

- Big mountain type (big wave type): it is composed of a large interval and a wide range of ups and downs. It is often associated with a certain kind of lofty feelings, broad singing, magnificent and heroic personality.

- Rising type: continuous upward melody often expresses high and agitated emotions.

- Drop type: continuously descending melody is often used to express emotions that change from tension to relaxation.

- Repeat type: the horizontal melody of homophonic repetition often plays a background role, or it is like retelling in an opera, to exaggerate its specific atmosphere.

- Stress type: the interrogative melody with the highest note at the end often produces a "questioning" effect similar to the questioning tone.

- Zigzag style: the melody has a small range of ups and downs, and the crest period is short, and it often fluctuates rapidly from a small interval in a not wide range. It can make the emotions appear vivid and lively.

- Surround type: surround sound moves up and down around the center tone.

The detection of waveform is slightly simplified in this paper. The difference sequence $D(a_n)$ we got above is judged by the positive and negative relationship. A positive number represents an increase in pitch, and a negative number represents a decrease in pitch. First, we start with the first number, check the following 3 numbers to see if they have the same sign, if they have the same sign, check the 3 numbers after the opposite sign to see whether the sign has changed, if there are no changes, the test round ends, and it is counted as a wave. Then start the test from the back.

Waviness test steps are as follows:

- The generated MIDI melody file is converted into a sequence $A_i$, $i = 1, 2, . . ., n$.

- Use $D_i = A_{i+1} - A_i$ to get sequence $D_i$.

- Starting from $i = 1$ cycle, view each element in $D_i$, validation of the current element behind the symbols and element symbols are the same, the same symbol illustrates three notes after the element is the same direction, then verify whether the element and the third element of symbols on the contrary, to verify whether the third and the fourth element with the same symbol, the same symbol means that the next three notes starting with the third note change in the same direction. And the direcyion is opposite to the first three.

- Set the initial value $k = 0$, and $k + 1$ if the above conditions are met.

Then compare the generated music with the number of waves in real music to make an evaluation.

*iii) Note-level mode test.* The note-level mode test is used to check whether the notes of the generated notes are all in the C major key we stipulated, from which the formula can be obtained:

$$d = \frac{Note_{in}}{Note_{all}} \tag{9}$$

$Note_{in}$ is the number of notes in the mode, and $Note_{all}$ is the number of all notes. The higher ratio of the notes falls on the mode to all notes, the generated melody is more consistant with the mode.

## Experiments

In this section, we introduce the selected data set and preprocess the data. After that we illustrate the parameters of the model set up and the computer configuration, and then generate the melody. The generated melody is analyzed through different starting characters, and then compared with the LSTM model. Finally, MEM is used to evaluate the generated melodies.

### Dataset

The data set we used is POP909 [30], and the music in this data set is 909 Chinese pop songs. we used the mid files in the folder and process it by using music21 [31] in Python. The music in this data set is clear, from which we can directly extract the music melody part through track selection, thus greatly reducing the time of data preprocessing, and making us focus on model training, evaluation, and construction.

### Preprocessing

All the songs were shifted to C major key. Then we process 909 popular songs through the above preprocessing method in IV to obtain a series of music melody texts. We divide the music texts into one hundred subsets to facilitate model training. In Fig 9, there are samples of resulting music text.

### Model training

We implemented our model using Tensorflow [32] and Keras [33] as' the backend. And used music21 [31] to process our data, The parameters are set as follows:

- The GPT-2 model used in the experiment is set as 8 hidden layers;

- Width of each layer is 1024;

- Dimension of word embedding is 64;

- Position vector was 1024;

- Number of multi-head attention mechanism is 4;

- Initial learning rate is 0.0001;

- Input data size is 1024;

- We used RTX2060 GPU for training, and the training rounds were set as 1000rounds.

After several rounds of experiments, it is found that the melody generated by the model trained with the above parameters is more in line with our auditory habits, and also has good

| 1 | G3 | A3 | ^ | ^ | ^ | C4 | ^ | D4 | E4 | ^ | ^ |
| 2 | E4 | ^ | D4 | E4 | ^ | ^ | ^ | D4 | E4 | - | - |
| 3 | G4 | ^ | G4 | ^ | ^ | G4 | - | - | - | G4 | G4 |
| 4 | B-4 | - | - | - | - | - | - | G#4 | - | - | - |
| 5 | G4 | - | - | - | C4 | ^ | ^ | D4 | - | - | - |
| 6 | E-4 | - | - | - | - | - | - | ^ | ^ | ^ | ^ |
| 7 | G3 | - | - | - | B-3 | ^ | C4 | - | - | - | ^ |
| 8 | E4 | ^ | ^ | G4 | G4 | ^ | ^ | G4 | ^ | G4 | ^ |
| 9 | E-4 | - | - | - | C4 | - | - | - | ^ | ^ | ^ |
| 10 | E3 | - | - | - | G3 | - | - | - | E3 | - | - |

**Fig 9. Sample music melody text after preprocessing, and each line represents the melody of a song.**

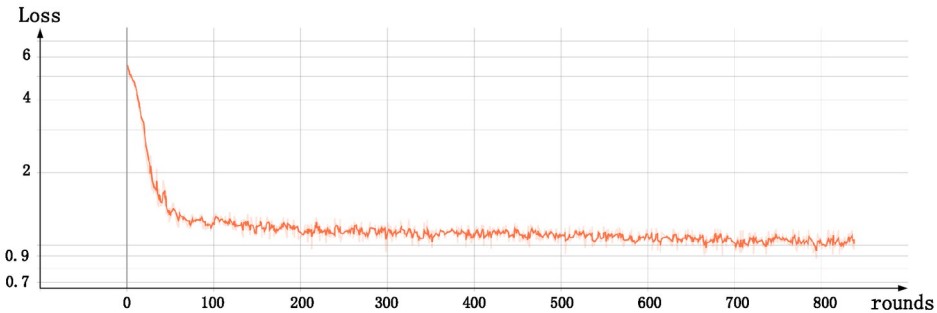

**Fig 10. The change of loss value during training, the abscissa is rounds and the ordinate is loss.**

results in the MEM model test.The change of loss with the number of training rounds in model training is shown in the Fig 10.

## Melody generation

Before generating the melody, specify the length of the notes to be generated and some parameters(length of generated notes, randomness of generated notes, etc.), then enter some driving characters $m_1, m_2, \ldots, m_n$.

When generating notes, we first input the initial notes. The length of this string of notes can be customized. Then set up to select one of the eight notes or symbols(eight notes or symbols with the highest probability predicted by the model) as the next one. MT-GPT-2 only generate one note at a time and the length is 1024. Converted to MIDI music at a speed of 90, the music lasts about 2 minutes and 30 seconds. The generated notes are converted into melody as shown in the Fig 11.

We designed three experiments to evaluate the generated music, divided into the following three parts: comparison with different initial notes, comparison with other models and evaluation using MEM. Through the comparison of different initiators, it can confirmed the performance of MT-GPT-2 and verify whether some characteristics of popular music melody have been learned. Compared with the LSTM model, it can be verified that the large pre-training model MT-GPT-2 can achieve better results in music generation. Finally, by MEM verification, we compared it with real music and LSTM model to verify whether the MT-GPT-2 model can meet the requirements of real music melody in a more objective perspective.

## Comparison with different initial notes

In order to verify whether the initial notes can guide the generation of melody, we use four different initial notes in Fig 9 to test the performance of the model.

As shown in Fig 11, the notes and rhythm in the red box are the notes we set in advance. The pitch and rhythm settings in (a) and (b) are more complicated, this is to test whether the model learned the musical characteristics of complex fragments. The pitch and rhythm settings in (c) and (d) are very simple, in order to detect the ability of the model to generate, and to test whether a relatively complex rhythm can be generated under a simple starting note melody.

In (a) and (b), the red box is the initial note. In the figure, we mark two blue boxes. The framed melody segment reflects the similar rhythm and pitch to the initial segment, but in the sound we can find the note and the rhythm are not exactly the same, the small changes are in line with the relative relationship between the bars in popular music.

The red frame in (c) and (d) is the initial note, which is a simple note and rhythm, but when the subsequent melody is generated, such as the green frame, you can see that it can

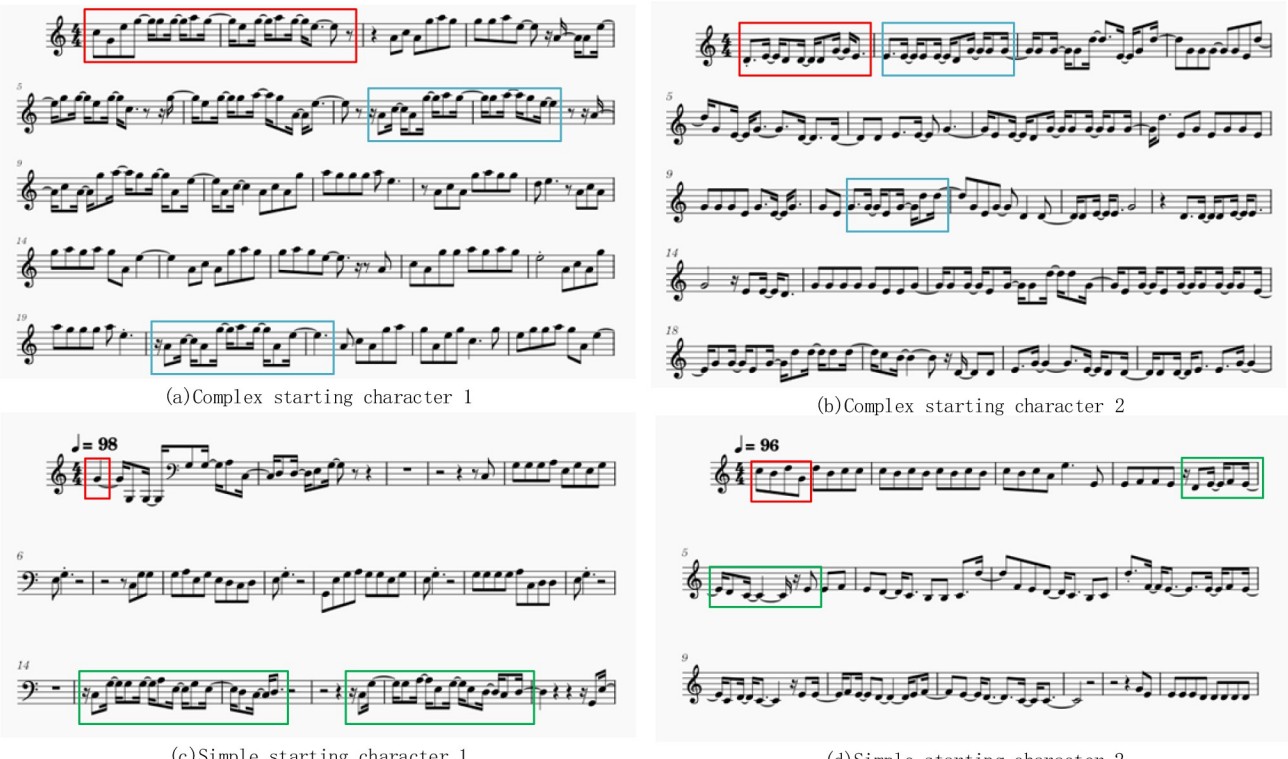

**Fig 11. Examples of using the MT-GPT-2 model to generate music, the note in red box is the starting note of the input, the notes framed in blue in (a) (b) are rhythmic pitches similar to the starting notes, while the notes framed in green in (c) (d) are rhythmic and pitch pitches unrelated to the starting notes.**

generate the relatively complex rhythmicity, which shows that the rhythmic text is useful in music generation.

From the music generated above, we can see that the music we generated has a series of characteristics of real music.

## Comparison with other models

In this section, we compare the LSTM model [6] with our MT-GPT-2 model, and use the same data set and data preprocessing method, where LSTM also has 8 layers and each layer has a width of 1024. For the training rounds, we set LSTM model of 500 rounds and MT-GPT-2 model and compared the training results of 1000 rounds. Taking into account the characteristic of musical melody, we analyzed the performance of the model from the aspect of musicology.

As Fig 12 shown, (a), (b) are generated by LSTM and (c), (d) are generated by MT-GPT-2, and the green square refers to each of the melody notes, next to the pitch for the piano keys. Red box labeled notes for the generated melody fragment is the tonic, blue braces mark out the pause part of the section. The note length of these (the length of the green square) corresponding to the rhythm of the segment, yellow boxes mark parts of the melody that are similar before and after, and pink arrows show the wave changes of the notes. When using the LSTM model, because of the limitation of the generated sequence length, the time interval between the tonics is too long, the LSTM model fails to pay good attention to the connection between the tonic and the before and after notes. Compared with the LSTM model, the MT-GPT-2 model

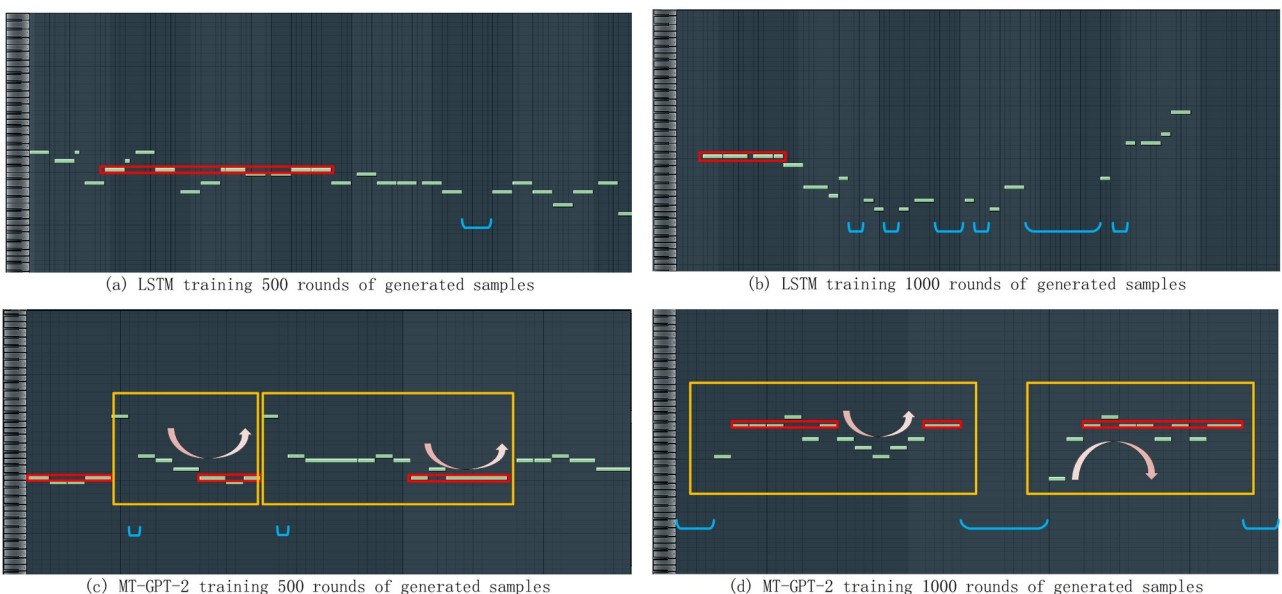

**Fig 12. MIDI files, (a) and (b) generated using the LSTM model were compared with MIDI files (c) and (d) generated using the MT-GPT-2 model.** (Use FL Studio DAW for presentation).

showed better performance for notes with a long interval, and could notice the tonic after several notes, and change around the tonic, which met the requirements of popular music creation.

In terms of rhythm, in the preceding (a) and (b), the rhythm formed by the duration of notes and pauses is somewhat disorganized, which does not have the characteristics of rhythm for the listener's hearing. However, in (c) and (d) the music generated by MT-GDT-2, it can be seen that the rhythm is relatively obvious, and there is a certain repetition, which is in line with the listeners' auditory habits.

For melody pitch change, we use the pink arrow in the figure to mark out the changing direction of the notes, in the (a) and (b) we can not see the intuitive waves of melody. The rhythm of (a) is downward all the time. The rhythm of (b) starts down and ends up rising, but the time intervals is too far, and the range of notes is too wide, it is not too conforms to the characteristics of popular music. The melodies in (c) and (d) are obviously wavy, and the rising and falling notes have a certain kinetic energy and the trend of change, which meets the requirements of pop music melody in music theory.

The music shown in (c) and (d) is generated by MT-GPT-2 model. The musical melodic fragments framed in yellow show that the melodies have obvious repeatability, which is a music characteristic that LSTM model can not generate.

All of these aspects of the analysis show that using a large pre-training model to generate music is better than some simple models.

## Evaluation using MEM

We use mathematical statistics evaluation method and music theory evaluation method (MEM) in IV to evaluate the quality of our model based on music textual melody generation.

**Mathematical statistics.** We use the MT-GPT-2 model and the LSTM model [6] to randomly generate many pieces of music, compare them with the real music in the data set, and take the average of each evaluation index as shown in Table 1.

**Table 1. Mathematical evaluation table.**

| Mathematical indicators | Whitney U test | Wilcoxon test | Kruskal-Wallis H test |
|---|---|---|---|
| MT-GPT-2 | **0.536371415** | **0.789986781** | **0.536285857** |
| Leak-GAN | 0.486661874 | 0.75105026 | 0.489012412 |
| Music SketchNet | 0.503457679 | 0.691353433 | 0.525464573 |
| LSTM | 0.099479186 | 0.672383992 | 0.099468169 |

Because the generate melodies is sometimes not on the same tone, sometimes differ octave and so on. In this paper, We subtract the previous pitch from the note's pitch to get a sequence of pitch variations so that we can not be restricted by the tonic and tonal to the music, comparing the similarities between such changes in series to eliminate the difference between the pitch of the tonic. Starting from the direction and size of the change of the notes, the comparison between the change and the real change music can be obtained through the above three ways of combing and testing.

We can see that among the various mathematical statistics, our model MT-GPT-2 model is closer to real music in terms of mathematical statistics. In the U test and K test, our model reached an evaluation of 0.5 or more. Compared with the LSTM model [6],Leak-GAN model [12] and Music SketchNet [13], our model has better results.

Table 1 shows the comparing results through traditional mathematical statistics methods. In the mathematical model, we use our new music representation method to convert the melody into a one-dimensional signal that greatly reduces the difficulty of music comparison and is very convenient when using comparison methods.

**Music theory statistics.** The smooth-saltatory progression comparison and wave test are creative in music evaluation. We use our new data representation to compare one-dimensional music melody signals to find out whether there is an objective gap between the generated melody and the real melody.

As for the smooth-saltatory progression comparison analysis, we make the following explanations to make people understand the meaning of this more intuitively.

In the Fig 13, the change degree between two notes in the blue frame is less than 4 intervals, which is called "smooth progression", and the change degree between notes in the red frame is more than 4 intervals, which is called "saltatory progression". The change of the notes in music will have a great impact on people's hearing feelings. The music is gradual, not high and low. When we listen to it, if there is a sudden change of more than 4 degrees, it will sometimes feel very obtrusive, so it is necessary to detect the change of notes.

As for the wave test analysis, we make the following explanations to make people understand the meaning of this more intuitively.

The melody generated in this section of the Fig 14, the notes under the blue arrow conform to our above rules. Three notes in a row show the same trend and then change in the opposite direction, which is called a wave. The wave shape test is an important aspect to identify whether a melody is smart. We can find that this kind of orderly, undulating changes is very

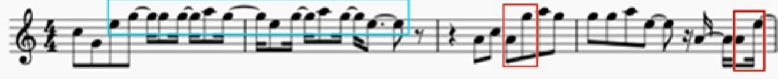

**Fig 13. Smooth-saltatory progression comparison analysis.**

**Fig 14. Wave test analysis.**

**Table 2. Music theory evaluation table.**

| Music Theory Index | smooth-saltatory progression comparison | Wave inspection | Tonality test |
|---|---|---|---|
| MT-GPT-2 | **0.277976884** | **145** | **1** |
| Leak-GAN | 0.237623762 | 134 | 0.8741215 |
| Music SketchNet | 0.257823489 | 167 | 0.9321346 |
| LSTM | 0.225806452 | 94 | 0.7171429 |
| Real Music | 0.207687974 | 123 | 1 |

common in a large number of famous songs. Our purpose here is to find out whether the model has learned this characteristic.

We use the MT-GPT-2 model and the LSTM [6] model to randomly generate ten pieces of music, compare them with the real music in the data set, and take the average of each evaluation index as shown in Table 2.

As shown in Table 2, in terms of the music theory of the generated music, we can see that the music generated by MT-GPT-2 model is very similar to real music in all aspects, and has more variability, and the mode is all in C major.

From the above Table 2, we can find that in our MT-GPT-2 model, the note changes and wave test are relatively active. This shows that our model has played a certain role in the learning of music. The effect of music generation is more musical than LSTM [6], Leak-GAN model [12] and Music SketchNet [13]. Secondly, our music is very accurate, all in the range of C major, so the generated melody is more in line with the aesthetic sense of hearing.

## Conclusion

In this paper, we described the MT-GPT-2 model, which is a GPT-2 music generation model based on music textual data. Thanks to this data preprocessing method, we have achieved consistency between text generation and music generation. The usage of special symbols to indicate the delay of the notes and the pause of the music is often overlooked in data processing.

The new textualized melodic approach we proposed is only one of many possible ways to represent it. We could have combined all the features in different ways (for example, separating information such as the pitch of a musical melody note and the length of a musical note, putting them in different vectors rather than together), and represented various information about music in different ways. We chose this approach because it seemed to be the most straightforward. In order to apply the large model to our music generation in a convenient and simple way, we have experimented with this method to ensure the integrity and representativeness of the music data.The purpose is to be able to use other large language models to quickly train and generate music.

At the same time, we have established a music melody evaluation model MEM, which is described through mathematical statistics and music theory. In this way we can evaluate music as objectively as possible and judge the quality of the model and the generated music.

Although the music evaluation method proposed by us can give us a more objective view of the model's ability to generate musical melodies, it does not reveal anything about the inner workings of the model. The interpretability of neural networks is a difficult problem, which is worth for further study.

Finally, the entire study was conducted in the context of MIDI data. However, this representation has some limitations. For example, the representation of some notes will be very long when the extraction of unit time is too small, resulting in the dilution of data. There is still no good solution for the text representation of multiple tracks. Studying the generation of chords under the melody and the generation of subsections will be the major research direction in the future.

## Author Contributions

**Formal analysis:** Yi Guo, Yangcheng Liu.

**Methodology:** Yangcheng Liu.

**Resources:** Yi Guo.

**Software:** Yangcheng Liu.

**Validation:** Yangcheng Liu.

**Visualization:** Yangcheng Liu.

**Writing – original draft:** Yangcheng Liu.

**Writing – review & editing:** Yi Guo, Ting Zhou, Liang Xu, Qianxue Zhang.

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
