## [Decision Letter · Decision Letter 0]

27 Jul 2022

PONE-D-22-12067An Automatic Music Generation and Evaluation Method Based on Transfer LearningPLOS ONE

Dear Dr. Zhou,

Thank you for submitting your manuscript to PLOS ONE. After careful consideration, we feel that it has merit but does not fully meet PLOS ONE’s publication criteria as it currently stands. Therefore, we invite you to submit a revised version of the manuscript that addresses the points raised during the review process.

The reviewers agree that the paper is promising but that the validation section is weak. The paper lacks a proper comparison with other methods of the state of the art. It is also required to specify the contribution of the paper, in terms of novelty. Finally, there are some presentation issues, such as figure reuse, that need to be addressed.

We look forward to receiving your revised manuscript.

Kind regards,

Marcelo Mendoza, Ph.D.

Academic Editor

PLOS ONE

Journal Requirements:

This research was supported by the Xihua University Graduate Innovation Fund (ycjj2020118), Open Research Fund of the Power System Wide-area Measurement and Control Sichuan Provincial Key Laboratory (2020JDS0027), Sichuan Science and Technology Program (2021YFG0337, 2020YFS0358, 2020YJ0367), the National Natural Science Foundation of China under Grant (61973257) and Grant SCITLAB-1021 of Intelligent Terminal Key Laboratory of SiChuan Province.

This research was supported by the Xihua University Graduate Innovation Fund 621

(ycjj2020118), Open Research Fund of the Power System Wide-area Measurement and 622

Control Sichuan Provincial Key Laboratory (2020JDS0027), Sichuan Science and 623

Technology Program (2021YFG0337, 2020YFS0358, 2020YJ0367), the National Natural 624

Science Foundation of China under Grant (61973257) and Grant SCITLAB-1021 of 625

Intelligent Terminal Key Laboratory of SiChuan Province

However, funding information should not appear in the Acknowledgments section or other areas of your manuscript. We will only publish funding information present in the Funding Statement section of the online submission form. 

This research was supported by the Xihua University Graduate Innovation Fund (ycjj2020118), Open Research Fund of the Power System Wide-area Measurement and Control Sichuan Provincial Key Laboratory (2020JDS0027), Sichuan Science and Technology Program (2021YFG0337, 2020YFS0358, 2020YJ0367), the National Natural Science Foundation of China under Grant (61973257) and Grant SCITLAB-1021 of Intelligent Terminal Key Laboratory of SiChuan Province.

Reviewers' comments:

Reviewer's Responses to Questions

**Comments to the Author**

1. Is the manuscript technically sound, and do the data support the conclusions?

Reviewer #1: Yes

Reviewer #2: Yes

2. Has the statistical analysis been performed appropriately and rigorously? 

Reviewer #1: Yes

Reviewer #2: Yes

3. Have the authors made all data underlying the findings in their manuscript fully available?

Reviewer #1: Yes

Reviewer #2: No

4. Is the manuscript presented in an intelligible fashion and written in standard English?

Reviewer #1: Yes

Reviewer #2: Yes

5. Review Comments to the Author

Reviewer #1: 1. major comment: the paper proposes a MT-GPT-model, under the pretraining of GPT language models, for melody generation. However, the texture representation it proposed has already been researched and explored in several papers of musicVAE, such us:

a. Adam Roberts, et al. "A Hierarchical Latent Vector Model for Learning Long-Term Structure in Music", ICML 2018.

b. Ke Chen, et al. "Music SketchNet: Controllable Music Generation via Factorized Representations of Pitch and Rhythm", ISMIR 2020.

c. Ziyu Wang, et al. "PIANOTREE VAE: Structured Representation Learning for Polyphonic Music", ISMIR 2020.

In that, this paper actually reuses an existing representation with an existing model, thus is of questionable novelty.

2. major comment: the paper only compares the proposed model with an LSTM model, with no reference to previous works. I suggest that the authors compare their model with (a) and (b) models, draw new conclusions and make the appropriate citations. Also, some other advanced models can also be added to compared using transformer models that are very similar to GPT:

d. Yu-Siang Huang, Yi-Hsuan Yang, "Pop Music Transformer: Beat-based Modeling and Generation of Expressive Pop Piano Compositions", ACM Multimedia 2020.

e. Wen-Yi Hsiao, Jen-Yu Liu, Yin-Cheng Yeh, Yi-Hsuan Yang, "Compound Word Transformer: Learning to Compose Full-Song Music over Dynamic Directed Hypergraphs", AAAI 2021.

3. major comment: the music evaluation metrics seems to be the main motivation and novelty of the paper, but it lacks examples to describe it better. For example, is there any example/demo for "smooth-saltatory progression comparison"?

4. minor comment: the paper reuses figures from other papers without proper permissions. For example, figure 9 is from the POP909 paper, I suggest removing it since it does not provide any additional information than directly demonstrating it in texts. Figure 12 needs to notate which song in pop909 it uses with a reference to the original track. Figure 2 and Figure 13 use the FL studio DAW, it should be footnoted. And many figures are poor resolution screenshots. I suggest that the author save it to higher DPI or vector figures.

Reviewer #2: The authors present a GPT2 model for automatic music generation and proposed a way to evaluate the generated music.

The paper seems well written, but with minor mistakes. (Line 36, "is", Line 50, "used", Line 55, "thirsty seems out of context", Line 92, "with", Line 136, "because", Line 350 and 351, space after punctuation)

In 193 capital P should be used.

First of all, there should be a shared repository and some samples of the generated music. (Github repository)

The experiments seem well performed, but the reviewer seems not be convinced by the objectivity of the valiudation method. They use several statistal tests, to present how different the generated music is from the original training data, but this doesn't mean that the generated music is good. It is only different. The musical theory tests seem well performed.

Also, it doesn't seem fair to compare a transformer model with only a LSTM. At least they should have compared it with another transformer model. distill-BERT, Performer, Linformer, Elektra, etc.

6. PLOS authors have the option to publish the peer review history of their article (what does this mean?). If published, this will include your full peer review and any attached files.

Reviewer #1: No

Reviewer #2: No

---

## [Author Response · Author response to Decision Letter 0]

11 Oct 2022

In response to a series of questions, I made the following changes to the article:

1. Modified the parts of the manuscript that do not conform to PLOS ONE style, including:

The acknowledgement section provided financial support but did not play any role in the study design, data collection and analysis. Therefore, the acknowledgement section was deleted as required and reflected in the Cover letter.

2. Revision based on the opinions of reviewers：

First of all, I would like to thank the reviewers for their criticism and correction. For their opinions, I hereby make the following modifications:

Reviewer 1:

Question:1. major comment: the paper proposes a MT-GPT-model, under the pretraining of GPT language models, for melody generation. However, the texture representation it proposed has already been researched and explored in several papers of musicVAE, such us:

a. Adam Roberts, et al. "A Hierarchical Latent Vector Model for Learning Long-Term Structure in Music", ICML 2018.

b. Ke Chen, et al. "Music SketchNet: Controllable Music Generation via Factorized Representations of Pitch and Rhythm", ISMIR 2020.

c. Ziyu Wang, et al. "PIANOTREE VAE: Structured Representation Learning for Polyphonic Music", ISMIR 2020.

In that, this paper actually reuses an existing representation with an existing model, thus is of questionable novelty.

Reply: According to the expert opinion, carefully read the above 3 articles. For the textual music in this paper, the differences in MIDI data processing methods are as follows: MIDI processing in this paper uses a more fine-grained music sampling method to extract music segments by taking the shortest note duration as the unit duration. For example, the music extraction in article a and b uses the customized frame number to extract music, which may cause the loss of music information. We've improved on that. Compared with MIDI music extraction in article c, which emphasizes the use of text to represent music, music extraction in c is divided into a tuple, which contains pitch and duration information, rather than one-dimensional signals. Therefore, it is described in Related Work in this paper.

Question: 2. major comment: the paper only compares the proposed model with an LSTM model, with no reference to previous works. I suggest that the authors compare their model with (a) and (b) models, draw new conclusions and make the appropriate citations. Also, some other advanced models can also be added to compared using transformer models that are very similar to GPT:

d. Yu-Siang Huang, Yi-Hsuan Yang, "Pop Music Transformer: Beat-based Modeling and Generation of Expressive Pop Piano Compositions", ACM Multimedia 2020.

e. Wen-Yi Hsiao, Jen-Yu Liu, Yin-Cheng Yeh, Yi-Hsuan Yang, "Compound Word Transformer: Learning to Compose Full-Song Music over Dynamic Directed Hypergraphs", AAAI 2021.

Reply: According to the opinions put forward by the experts, I carefully studied the above papers, and I also studied many music generation models before. Because in the field of NLG, the needs of text generation models and rapid development, the purpose of this article is to build a method that facilitates other large text generation models directly to generate music. Not only can you use the GPT model, but you can also use other languages to generate models such as GAN and Bert such as GAN and Bert. Compared with the LSTM model, it is to show the superiority of the use of large language models to generate music, which can generate longer and better music, indicating that using large language models to generate music is feasible and developed. For articles D and E, because the training data used is different, the number of music tracks is different, and the music is more difficult and unfair. Music generation cannot only be more accurate and loser, but requires comparison and pleasant, and this is the most difficult.

According to experts' suggestions, this article uses the same dataset and method of Music SketchNet models and Leak-GAN models for training and comparison. As shown in the article.

Question:3. major comment: the music evaluation metrics seems to be the main motivation and novelty of the paper, but it lacks examples to describe it better. For example, is there any example/demo for "smooth-saltatory progression comparison"?

Reply: As for the music evaluation model I built, due to space limitation, I could not show specific examples and demonstrations. However, the demonstration and related descriptions are presented in FIG. 13 on page 18, and the formula principle is also given in the article.

Question:4. minor comment: the paper reuses figures from other papers without proper permissions. For example, figure 9 is from the POP909 paper, I suggest removing it since it does not provide any additional information than directly demonstrating it in texts. Figure 12 needs to notate which song in pop909 it uses with a reference to the original track. Figure 2 and Figure 13 use the FL studio DAW, it should be footnoted. And many figures are poor resolution screenshots. I suggest that the author save it to higher DPI or vector figures.

Reply: Irrelevant images were removed in the article, and for Figure 12, the tracks in POP909 were not used. Instead, use the tracks generated by the model in this article. Footnotes have been added to the corresponding pictures.

Reviewer 2:

Question:The authors present a GPT2 model for automatic music generation and proposed a way to evaluate the generated music.

The paper seems well written, but with minor mistakes. (Line 36, "is", Line 50, "used", Line 55, "thirsty seems out of context", Line 92, "with", Line 136, "because", Line 350 and 351, space after punctuation)

In 193 capital P should be used.

Reply: Thank you very much for reading my article carefully. I have made corresponding modifications one by one in view of the language problems mentioned.

Question: First of all, there should be a shared repository and some samples of the generated music. (Github repository)

The experiments seem well performed, but the reviewer seems not be convinced by the objectivity of the valiudation method. They use several statistal tests, to present how different the generated music is from the original training data, but this doesn't mean that the generated music is good. It is only different. The musical theory tests seem well performed.

Also, it doesn't seem fair to compare a transformer model with only a LSTM. At least they should have compared it with another transformer model. distill-BERT, Performer, Linformer, Elektra, etc.

Reply: Thanks to the expert's opinion, first of all, the evaluation using statistical methods in order to confirm whether the generated music is close to the real music in the distribution in terms of mathematical statistical probability, which can show the quality of the training model to a certain extent. Using music theory evaluation can indicate how good or bad the generated music is, it's different, but I think it's all necessary. To solve the problem that there are few models to compare, I added leak-GAN and Music SketchNet in this paper for comparison.

---

## [Decision Letter · Decision Letter 1]

2 Mar 2023

An Automatic Music Generation and Evaluation Method Based on Transfer Learning

PONE-D-22-12067R1

Dear Dr. Zhou,

We’re pleased to inform you that your manuscript has been judged scientifically suitable for publication and will be formally accepted for publication once it meets all outstanding technical requirements.

Kind regards,

Marcelo Mendoza, Ph.D.

Academic Editor

PLOS ONE

Additional Editor Comments (optional):

Both reviewers agreed that this version is an improved version of the original draft and the comments have been succesfully addressed in this new version. Thus, I recommend the acceptance of your paper. Congratulations.

In your definitve version, please take care of this question formulated by one of the reviewers: Regarding the note multiple Ni in the chapter of extracting note length information, if the tempo of a song changes from 44 beats to 43 beats in the middle, Ni may not be an integer at this time, how do you deal with it? PLease take care also of this comment: Regarding references, please carefully review the bibliography section. Several references have incomplete metadata. Please conduct a thorough review of the reference list including all available metadata. Conference papers must include name of conference, publisher, places, dates, pages in proceedings. References to journal articles should include pages, volume, number, publisher. As for the list of authors, use the same convention. In your list, some references use first name, last name, others last name, initial.

Reviewers' comments:

Reviewer's Responses to Questions

**Comments to the Author**

1. If the authors have adequately addressed your comments raised in a previous round of review and you feel that this manuscript is now acceptable for publication, you may indicate that here to bypass the “Comments to the Author” section, enter your conflict of interest statement in the “Confidential to Editor” section, and submit your "Accept" recommendation.

Reviewer #3: All comments have been addressed

Reviewer #4: (No Response)

2. Is the manuscript technically sound, and do the data support the conclusions?

Reviewer #3: Yes

Reviewer #4: (No Response)

3. Has the statistical analysis been performed appropriately and rigorously? 

Reviewer #3: Yes

Reviewer #4: (No Response)

4. Have the authors made all data underlying the findings in their manuscript fully available?

Reviewer #3: Yes

Reviewer #4: (No Response)

5. Is the manuscript presented in an intelligible fashion and written in standard English?

Reviewer #3: Yes

Reviewer #4: (No Response)

6. Review Comments to the Author

Reviewer #3: The authors have adequately argued the comments made during the review. Some of these comments have been incorporated based on new experiments and others in the response letter based on argumentation that justifies the decisions made in the paper. I think the authors have done a thorough job addressing the observations.

Regarding references, please carefully review the bibliography section. Several references have incomplete metadata. Please conduct a thorough review of the reference list including all available metadata. Conference papers must include name of conference, publisher, places, dates, pages in proceedings. References to journal articles should include pages, volume, number, publisher. As for the list of authors, use the same convention. In your list, some references use first name, last name, others last name, initial.

Reviewer #4: 1. Regarding the note multiple Ni in the chapter of extracting note length information, if the tempo of a song changes from 44 beats to 43 beats in the middle, Ni may not be an integer at this time, how do you deal with it?

7. PLOS authors have the option to publish the peer review history of their article (what does this mean?). If published, this will include your full peer review and any attached files.

Reviewer #3: No

Reviewer #4: No

---

## [Editor Report · Acceptance letter]

17 Apr 2023

PONE-D-22-12067R1 

An Automatic Music Generation and Evaluation Method Based on Transfer Learning 

Dear Dr. Zhou:

I'm pleased to inform you that your manuscript has been deemed suitable for publication in PLOS ONE. Congratulations! Your manuscript is now with our production department. 

Kind regards, 

on behalf of

Dr. Marcelo Mendoza 

Academic Editor

PLOS ONE